# Monomeric CXCL12-Engineered Adipose-Derived Stem Cells Transplantation for the Treatment of Ischemic Stroke

**DOI:** 10.3390/ijms25020792

**Published:** 2024-01-08

**Authors:** Haoran Zheng, Khan Haroon, Mengdi Liu, Xiaowen Hu, Qun Xu, Yaohui Tang, Yongting Wang, Guo-Yuan Yang, Zhijun Zhang

**Affiliations:** 1Shanghai Jiao Tong University Affiliated Sixth People’s Hospital, School of Biomedical Engineering, Shanghai Jiao Tong University, Shanghai 200030, China; horan_zheng@163.com (H.Z.); haroonkhan8888@hotmail.com (K.H.); liumandy2022@163.com (M.L.); huxiaowen1106@163.com (X.H.); yaohuitang@163.com (Y.T.); yongting.wang@gmail.com (Y.W.); 2Health Management Center, Department of Neurology, Renji Hospital of Medical School of Shanghai Jiao Tong University, Shanghai 200127, China; xuqun@renji.com

**Keywords:** monomeric CXCL12, stroke, adipose derived stem cells, CXCR4, AMD3100

## Abstract

Adipose-derived stem cells (ASCs) possess therapeutic potential for ischemic brain injury, and the chemokine CXCL12 has been shown to enhance their functional properties. However, the cumulative effects of ASCs when combined with various structures of CXCL12 on ischemic stroke and its underlying molecular mechanisms remain unclear. In this study, we genetically engineered mouse adipose-derived ASCs with CXCL12 variants and transplanted them to the infarct region in a mice transient middle cerebral artery occlusion (tMCAO) model of stroke. We subsequently compared the post-ischemic stroke efficacy of ASC-mCXCL12 with ASC-dCXCL12, ASC-wtCXCL12, and unmodified ASCs. Neurobehavior recovery was assessed using modified neurological severity scores, the hanging wire test, and the elevated body swing test. Changes at the tissue level were evaluated through cresyl violet and immunofluorescent staining, while molecular level alterations were examined via Western blot and real-time PCR. The results of the modified neurological severity score and cresyl violet staining indicated that both ASC-mCXCL12 and ASC-dCXCL12 treatment enhanced neurobehavioral recovery and mitigated brain atrophy at the third and fifth weeks post-tMCAO. Additionally, we observed that ASC-mCXCL12 and ASC-dCXCL12 promoted angiogenesis and neurogenesis, accompanied by an increased expression of bFGF and VEGF in the peri-infarct area of the brain. Notably, in the third week after tMCAO, the ASC-mCXCL12 exhibited superior outcomes compared to ASC-dCXCL12. However, when treated with the CXCR4 antagonist AMD3100, the beneficial effects of ASC-mCXCL12 were reversed. The AMD3100-treated group demonstrated worsened neurological function, aggravated edema volume, and brain atrophy. This outcome is likely attributed to the interaction of monomeric CXCL12 with CXCR4, which regulates the recruitment of bFGF and VEGF. This study introduces an innovative approach to enhance the therapeutic potential of ASCs in treating ischemic stroke by genetically engineering them with the monomeric structure of CXCL12.

## 1. Introduction

Stroke is the second-leading cause of death and the third-leading cause of neurological disabilities worldwide, with limited treatment opportunities [1,2,3]. At present, the tissue plasminogen activator (t-PA) is the only FDA-approved thrombolytic drug that benefits less than 5% of stroke patients [4,5]. However, these treatments have a narrow therapeutic window and carry the risk of intracranial hemorrhage [6,7], which largely limits their clinical use. Consequently, there is an urgent need to expand the therapeutic window and reduce the subacute phase complication in ischemic stroke. The administration of stem cells during the subacute phase of stroke holds the potential to offer greater benefits to a wider spectrum of patients by mitigating initial-phase, stroke-related injuries, thus reducing subsequent complications arising from secondary damage [8,9].

Adipose-derived stem cells (ASCs) isolated from adipose tissue are multipotent mesenchymal stem cells (MSCs) [10]. Adipose tissue is both abundant and easily obtainable in large quantities [11]. Recently, ASC therapy has gained significant attention for its potential to facilitate the repair of various organs [12]. ASCs have been incorporated into human trials for autoimmune diseases such as multiple sclerosis, polymyositis, dermatomyositis, and rheumatoid arthritis [13]. Thus, ASCs present a promising alternative source of stem cells for application in mesenchymal tissue-based regeneration and engineering.

The CXCL12/CXCR4 axis plays a critical role in various functions, including promoting the homing of hematopoietic stem cells and EPC-mediated neuroprotection [14]. CXCR4 can form dimers with CXCL12, enhancing chemotaxis through CXCL12. Circulating CXCL12 exists in monomeric or dimeric form depending on its concentration. Dimeric CXCL12 has been explored in cancer immunology, where it suppresses T cell motility [15]. On the other hand, monomeric CXCL12 activates cell migratory responses and promotes the function of human endothelial progenitor cells in vitro [16,17]. However, the specific mechanisms underlying how monomeric and dimeric CXCL12 interact with their receptors and regulate signaling pathways remain unclear.

The potential of the monomeric structure of CXCL12 in transducing distinct cellular signals and effects as a potential treatment of stroke remains to be explored. Additionally, strategies to achieve multitarget and multi-mechanism modulation within the intricate and dynamic inflammatory environment of the brain through ASCs are currently lacking. To address these challenges, we engineered ASCs with CXCL12 variants and aimed to generate a synergistic effect in improving ischemic stroke outcomes.

In this study, we stereotactically injected ASCs-mCXCL12, ASCs-dCXCL12, ASCs-wtCXCL12, and ASCs into the peri-infarct area of mice one-week following tMCAO. Our findings demonstrate that treatment with ASCs-mCXCL12 relieved the mice’s brain ischemia more effectively. This was evidenced by enhanced functional recovery, increased angiogenesis, and neurogenesis after an acute ischemic stroke in mice.

## 2. Results

### 2.1. Identification of ASCs’ and CXCL12/Variants’ Gene Overexpression

Primary ASCs were isolated from subcutaneous white fat tissue (WAT) in the bilateral inguinal area of ICR mice (Male, 6 weeks). To identify ASCs, we performed flow cytometry to detect the expression of each cluster of each differentiation (CD) marker of ASCs. The result indicated that, in the second passage (P2) ASCs, 95.1% were CD29^+^, 99.8% were CD44^+^, 4.43% were CD34^+^, and 0.92% were CD45^+^. These findings were consistent with previous studies (Figure 1A).

For the overexpression of CXCL12 variants, we used lentiviruses carrying CXCL12 variants to transduce ASCs at 60% confluency at the third passage. The transduction efficiency was 85% based on the quantification of GFP-positive ASCs (Figure 1B). To further validate the expression of CXCL12, we performed Western blot and real-time PCR analysis. The results showed that CXCL12 was significantly increased in ASC-wtCXCL12, ASC-dCXCL12, and ASC-mCXCL12 compared to both the ASC and ASC-GFP groups (*p* < 0.05). Interestingly, different structures of CXCL12 did not affect its overall expression level (Figure 1C).

### 2.2. ASC-mCXCL12 Transplantation Improved Neurobehavioral Recovery and Reduced Brain Atrophy in Ischemic Mice

To investigate the impact of CXCL12 and its variants on the therapeutic efficacy of ASCs in tMCAO mice, we conducted a 5-week experiment (Figure 2A). Neurobehavioral tests were employed to assess the effect of ASCs with CXCL12 variants on the recovery of neurological function (Figure 2B). The mNSS test results showed that the neurological deficits decreased in the ASC and ASC-CXCL12 variant groups from the 3 weeks after tMCAO compared to the control group (*p* < 0.05). Additionally, the ASC-mCXCL12 group exhibited the most significant improvement among the therapeutic groups (*p* < 0.05). The rotarod test results showed significant improvement in motor function for all therapeutic groups at 5 weeks compared to the control group. Similarly, the hanging wire test and EBST test indicated higher average scores for the treatment groups 3 weeks after tMCAO, suggesting that ASC and ASC-CXCL12 variants could enhance the neuromotor and coordination function.

To confirm whether the ASC and ASC-CXCL12 variants’ transplantation could protect against brain injury, cresyl violet staining was performed 5 weeks after tMCAO. The results demonstrated that both ASC and ASC-CXCL12 variants reduced brain atrophy volumes compared to the control group (*p* < 0.05, Figure 2C). Notably, ASC-mCXCL12 showed a significant decrease in brain atrophy volumes compared to both ASC-wtCXL12 and ASC-dCXCL12 (*p* < 0.05), indicating that mCXCL12 confers improved benefits of ASC transplantation in tMCAO mice compared to wtCXCL12 and dCXCL12.

### 2.3. ASC-mCXCL12 Transplantation Improved Angiogenesis of Ischemic Mice

To check the state of ASCs 3 weeks after tMCAO, we detected GFP signals in ipsilateral brain slices from the ASC-GFP group and found no green signals in the peri-infarct areas (Figure 3A). The absence of a GFP signal suggests that the transplanted ASCs may perform the therapeutic effects by releasing signaling factors. To assess the blood vessel density after treatment, we conducted immunostaining of blood vessel marker CD31 5 weeks after tMCAO. Ki67, a marker of newly proliferated endothelial cells, was detected together with CD31 (Figure 3B). The result revealed a significant increase in blood vessel intensity in ASC and ASC-CXCL12 variant groups compared to the control group (*p* < 0.05, Figure 3C). Furthermore, the ratio of blood vessel intensity in the ipsilateral-to-contralateral hemisphere was higher than in the ASC and ASC-CXCL12 variant groups (Figure 3D). The presence of a higher number of CD31^+^/Ki67^+^ blood vessels in ASCs and ASCs-CXCL12 variants compared to the control group suggests that ASC, ASC-wtCXCL12, ASC-dCXCL12, and ASC-mCXCL12 not only increase blood vessel density but also promote angiogenesis in tMCAO mice (Figure 3E).

### 2.4. ASCs-mCXCL12 Promoted the Expression of VEGF and bFGF after Ischemic Stroke

VEGF and bFGF are key factors for angiogenesis post-injury [18]. To elucidate the underlying molecular mechanism responsible for better outcomes in the ASCs-mCXCL12 group, we employed Western blot and real-time PCR analysis. The quantification of VEGF and bFGF expression revealed that ASC-mCXCL12 transplantation promoted VEGF and bFGF levels in the peri-infract area of the tMCAO mice (*p* < 0.05, Figure 4A). Consistent with the Western blot result, real-time PCR tests for VEGF and bFGF expression in tMCAO mice showed that ASC-mCXCL12 treatment upregulated the VEGF and bFGF expression (Figure 4B). To explore the CXCL12 overexpression, we assessed CXCL12 expression via real-time PCR. The result showed that CXCL12 expression was significantly increased in ASC-wtCXCL12, ASC-dCXCL12, ASC-mCXCL12, and ASC groups (*p* < 0.05). Furthermore, we noted that the increased CXCL12 expression exhibited no significant differences among the therapeutic groups, whereas CXCR4 expression was increased in the ASC-mCXCL12 group (*p* < 0.05, Figure 4C).

### 2.5. ASC-mCXCL12 Transplantation Promoted Angiogenesis and Neurogenesis in Ischemic Mouse Brain

Based on our previous results, which showed that ASC-mCXCL12 significantly improved neuromotor and coordination function in ischemic mice from the third week after tMCAO and achieved a better therapeutic outcome, we conducted a series of experiments to highlight the benefits of ASC-mCXCL12 in comparison to PBS, ASC, ASC-wtCXCL12 and ASC-mCXCL12 at the 3 weeks after tMCAO. We designed the experimental schedule as shown in Figure 5A. Cresyl violet staining showed that ASC-mCXCL12 significantly reduced brain atrophy volumes compared to other groups (Figure 5B). H&E staining results showed that the ADSC-mCXCL12 group had a better structure recovery compared to the other three groups. Fluorescent staining of blood vessel intensity and CD31^+^/Ki67^+^ endothelial cells demonstrated that the ASC-mCXCL12 group increased newly proliferated vessels. To analyze whether ASC-mCXCL12 promoted neurogenesis in the peri-infarct area, we performed double immunostaining of DCX (Doublecortin) and Ki67. The results indicated a significant increase in DCX^+^/Ki67^+^ cells in the SVZ of the ASC-mCXCL12 group (*p* < 0.05, Figure 5C). Further analysis using Western blot (Figure 5D) and real-time PCR (Figure 5E) was conducted to detect the expression of VEGF and bFGF. The results showed that VEGF and bFGF expression was notably increased in the ASC-mCXCL12 group, which signifies the promotion of angiogenesis and neurogenesis in the mouse brain 3 weeks after tMCAO.

### 2.6. The Regulation of ASC-mCXCL12 Functions in Ischemic Mice Is Mediated Using the CXCL12/CXCR4 Signaling Axis

To verify whether CXCR4 affects neurological recovery in tMCAO mice undergoing ASC-mCXCL12 treatment, we administered ASC-mCXCL12 along with CXCR4 antagonist (AMD3100), PBS, ASC, and ASC-mCXCL12 in the tMCAO mice for a duration of 3 weeks (Figure 6A). The mNSS test results demonstrated that the AMD3100-injected group showed increased neurological deficits compared to the ASC-mCXCL12 group. Rotarod test results revealed that mice in the AMD3100 group experienced difficulties in maintaining balance and had significantly lower average scores compared to ASC-mCXCL12 in the hanging wire and EBST tests (*p* < 0.05, Figure 6B). The cresyl violet staining assay showed that the AMD3100 group exhibited increased atrophy volumes (Figure 6C). H&E staining of the ASC-mCXCL12-AMD3100 group showed worse structure recovery than the ASC-mCXCL12 group. In comparison to the ASC-mCXCL12-AMD3100 group, the ASC-mCXCL12 group exhibited higher blood vessel intensity, a large amount of newly formed capillaries, and an increased level of DCX^+^/Ki67^+^ cells in SVZ (Figure 6D). These results suggest that AMD3100 regulates ASC-mCXCL12 functions by inhibiting CXCR4 binding with CXCL12. Western blot and real-time PCR analyses further demonstrated that, in the presence of AMD3100, ASC-mCXCL12 transplantation did not upregulate VEGF and bFGF expression in the peri-infarct area after ischemia (Figure 6E,F).

## 3. Discussion

In this study, we harnessed lentiviral vectors to introduce monomeric, dimeric, or wild-type CXCL12 genes into ASC. Our results demonstrated that stereotactic ASC-mCXCL12 transplantation successfully promoted neurobehavioral recovery by stimulating angiogenesis and neurogenesis. Overall, our findings present a novel approach for the engineering of ASC, highlighting the potential of utilizing monomeric CXCL12 to enhance paracrine mechanisms for the treatment of stroke (Figure 7).

ASCs play a pivotal role in stimulating angiogenesis through the secretion of VEGF, suggesting their potential in wound healing [19,20,21]. Previous studies have shown that ASC transplantation could alleviate neurological functional deficits, decrease infarct volume and brain atrophy, and reduce apoptosis and inflammation in cerebral vascular disease [22,23,24]. However, the low survival rate of transplanted ASCs [25] suggests that alternative strategies are needed to promote ASCs’ paracrine effect. As supported by numerous reports, ASCs exhibit the capacity to secrete a diverse secretome, which contributes to cell proliferation, differentiation, migration, and enhancements in cellular and microenvironment protection [26,27,28,29]. This secretome corresponds to a panel of trophic factors, such as cytokines, growth factors, and chemokines, which allow ASCs to act as paracrine mediators that are more effective than cell replacements. Thus, we adopted virus-carrying variant genes of CXCL12 to transfect ASCs and analyzed a wider treatment time window for ischemic stroke.

In our study, we observed a significant increase in CXCR4 expression in the ASC-dCXCL12, ASC-mCXCL12, and ASC-wtCXCL12 groups. The ASC-mCXCL12 group showed higher expression of CXCR4, suggesting the potential existence of a positive feedback loop regulating CXCR4 expression. CXCR4 is widely recognized as a crucial receptor that is responsible for initiating signaling cascades upon interaction with CXCL12 [30]. Upon monomeric CXCL12 binding to CXCR4, CXCR4 triggers the dissociation of the heterotrimeric G protein into Gα and Gβγ subunits and converts guanosine diphosphate bound to the G protein to guanosine triphosphate, followed by the activation of downstream signaling [31]. The Gβγ subunit activates PLC, which results in the conversion of PIP2 to DAG and IP3 and the release of Ca^2+^, followed by PKC activation and the phosphorylation of target proteins. Gαi and Gβγ subunits activate the PI3K-AKt pathway. Subsequent AKt activation regulates gene expression [32]. Generally, monomeric CXCL12 induced the mobilization of intracellular calcium, suppressed cAMP signaling, facilitated the recruitment of β-arrestin-2, promoted the accumulation of filamentous actin, and enhanced cell migration. It was observed that specific interactions between CXCL12 monomers and CXCR4 were compromised upon dimerization [16]. Notably, CXCL12 does not trigger signaling upon binding to CXCR7; instead, it facilitates signaling by forming functional heterodimers with CXCR4 [33]. Moreover, its agonists can lead to the down-regulation of CXCR4 and, consequently, inhibit the functionality of CXCL12 [34]. CXCR7 acts as an auxiliary receptor and regulator in chemokine signaling [17]. While CXCR4 is highly expressed in many cell types such as lymphocytes, endothelial cells, and hematopoietic stem cells [35], recent studies have demonstrated the essential role of the CXCR4/CXCL12 axis in tissue repair, encompassing critical processes such as hematopoiesis, organogenesis, developmental processes, vascularization, carcinogenesis, and neurogenesis [36,37,38]. The pathway monomeric CXCL12-CXCR4-PI3K-Akt in our study would be activated and promote gene transcription, like VEGF and bFGF. Then, these expressed genes could sufficiently increase angiogenesis and neurogenesis in the brain to influence neurobehavioral recovery.

Based on our previous research, we found that AMD3100 primarily functions as a CXCR4 antagonist, thereby indicating that the CXCL12/CXCR4 pathway holds significant potential as a promising target for drug development [39]. In our current study, the results demonstrated that AMD3100 injection reverses the beneficial effect observed in angiogenesis and neurogenesis within the ASC-mCXCL12 group. The AMD3100 injection group also exhibited increased neurological functional deficits that aggravated edema volume and brain atrophy. These findings strongly suggest that monomeric CXCL12 interacts with CXCR4 to facilitate neurobehavior recovery after ischemic stroke.

Currently, in clinical practice, only the tissue plasminogen activator (tPA) and thrombectomy have demonstrated efficacy in treating ischemic stroke [40,41]. Therefore, we present a novel strategy involving the utilization of the monomeric structure of CXCL12 to enhance the paracrine effect of ASCs. In the field of ischemic stroke neurorepair, angiogenesis and neurogenesis are essential processes and serve as essential indicators for assessing the effectiveness of therapeutic interventions [42]. Recently developed capillaries play a crucial role in delivering oxygen and nutrients to the ischemic region, thereby promoting tissue repair and remodeling. Similarly, newly generated neuroblasts possess the ability to migrate to the damaged areas, where they mature into neurons, effectively replacing the dead neurons [43]. Our study has demonstrated that the transplantation of ASCs engineered with CXCL12 monomers results in the upregulation of VEGF and bFGF expression in the mouse brain following an ischemic stroke. This, in turn, could facilitate angiogenesis and neurogenesis, ultimately leading to improved neurobehavioral recovery.

While our study has demonstrated that engineered ASCs could promote angiogenesis and neurogenesis, and that ASCs have a lower propensity for tumor formation compared to other cell types [44,45], there are long-term effects and potential risks, including the possibility of cancer after ASC transplantation beyond the 5 weeks required for a comprehensive investigation. Second, recent research has revealed that receptor ACKR1 exhibits a distinct inclination towards forming stronger associations with CXCL12 dimer [46], a factor that was not investigated in our study. Additionally, while our study shows the interaction between mCXCL12 and CXCR4, it is important to acknowledge that ASC transplantation has been linked to the recruitment of VEGF and bFGF in injury lesions. Further investigation is required to explore whether mCXCL12 transplantation exclusively interacts with CXCR4, leading to the elevated expression of VEGF and bFGF [47,48]. Our study offers valuable insights into the treatment of ischemic stroke in rodents, and It is essential to acknowledge that the use of mouse-derived ASCs may limit the direct clinical translation of our findings to human therapy. Moreover, a limited sample size, potential variability in individual ASC responses, and the complexity of immune system responses are some factors that may affect our conclusions.

## 4. Materials and Methods

### 4.1. Experimental Design

Animal experiments were carefully conducted in accordance with the Animal Research: Reporting of In Vivo Experiments (ARRIVE) guidelines, as reported by [49], and received approval from the Institutional Animal Care and Use Committee of Shanghai Jiao Tong University, Shanghai, China. A total of 130 adult male ICR mice, aged 6–8 weeks and weighing approximately 25 ± 3 g, were selected for use in the controlled cortical impact (CCI) surgery. These mice were procured from Vital River Laboratory Animal Technology, Beijing, China, under license No. SCXK (Jing) 2019-0001.

In order to mitigate the influence of varying hormone levels [50], male mice were chosen for the experimental procedures. The mice were kept in the animal care facility at Shanghai Jiao Tong University, where they experienced a consistent 12/12 h light/dark cycle and a steady temperature of 24 °C. Humidity levels were maintained within the range of between 40% and 70%, and the mice had unrestricted availability of food and water. Each cage housed four mice, and the allocation of animals to specific treatment groups was carried out through random selection. For the 5-week experiment, the sample sizes were as follows: PBS Group (*n* = 10), ASCs Group (*n* = 10), ASCs-wtCXCL12 (*n* = 10), ASCs-dCXCL12 (*n* = 10), ASCs-mCXCL12 (*n* = 10). We used at least 5–6 mice to obtain the brain slices and a minimum of 3 mice for the brain tissues. At the 3-week timepoint, the animals were assigned as follows: PBS Group (*n* = 10), ASCs Group (*n* = 10), ASCs-wtCXCL12 (*n* = 10), ASCs-mCXCL12 (*n* = 10), ASCs-mCXCL12-AMD3100 (*n* = 10). In this study, we used 100 ICR male mice for the tMCAO model; 11 mice died during the experiment, and the mortality rate was 11%. The mice were trained on a rotarod machine before transient middle cerebral artery occlusion (tMCAO). Neurobehavioral tests were performed at 7, 14, 21, 28, and 35 days after tMCAO, and the animals were euthanized at 3 weeks and 5 weeks after tMCAO according to the experimental design.

### 4.2. PLenti-CXCL12-IRES-GFP Vector Construction and Lentiviral Vector Production

The PLenti-CXCL12-IRES-GFP vector design involved integrating mouse CXCL12α cDNA into the multi-cloning site of the pLenti-IRES-GFP platform. Furthermore, the pLenti-CXCL12L36C/A65C construct was created to facilitate the dimerization of CXCL12 following the Q5^®^ Site-Directed Mutagenesis Kit Quick Protocol (New England Biolabs, Ipswich, MA, USA) [51]. The pLenti-CXCL12L55C/I58C construct was used as a representation of monomeric CXCL12 [52].

To generate lentiviruses carrying CXCL12 monomer (LV-mCXCL12), CXCL12 dimer (LV-dCXCL12), and wild-type CXCL12 (LV-wtCXCL12), the respective CXCL12 plasmids or variants were co-transfected with pCMV-VSV-G and pCMV-delta plasmids into 293T cells.

### 4.3. ASCs Isolation and Characterization

We isolated the ASCs from 6-week-old male ICR mice. Briefly, the inguinal subcutaneous adipose tissues were carefully extracted from the mice. Then, the isolated adipose tissues were manually sliced and then digested with 0.1% type II collagenase (Sigma–Aldrich, Saint Louis, MO, USA). The isolation process was carried out at 37 °C for a duration of 30 min, under shaking conditions of 20 rpm. Digestion of sliced adipose tissues was stopped by adding an equivalent volume of low glucose medium (L-DMEM, HyClone, Logan, UT, USA) containing 10% fetal bovine serum (FBS) (Gibco, Carlsbad, CA, USA). Subsequently, the solution was filtered through a 70 μm filter and centrifuged for 5 min at 300 g. The cells were re-suspended in a complete medium and cultured at 37 °C in a 5% carbon dioxide environment with full humidity saturation. The culturing medium was prepared from L-DMEM supplemented with 10% FBS and 1% penicillin/streptomycin (Invitrogen, Carlsbad, CA, USA) and was renewed every two to three days. Cells were subcultured using 0.25% trypsin–EDTA. Cells between passages 2 and 5 were collected and used for all the experimental studies. Primary ASCs were used and subcultured each time. Flow cytometry was employed to validate the ASCs’ surface markers (Accuri C6, BD, Franklin Lakes, NJ, USA). The isolated single-cell suspension underwent three washes with PBS supplemented with 1% BSA. Subsequently, the cells’ suspension was incubated in the dark at 4 °C for 30 min with specific murine antibodies, namely, anti-CD29-PE (Cat#562801, BD), anti-CD34-Alexa Fluor 647 (Cat#560233, BD), anti-CD44-APC (Cat#561862, BD), and anti-CD45-PerCP (Cat#561047, BD). As a control, we used a cell suspension not incubated with antibodies.

### 4.4. Transient Middle Cerebral Artery Occlusion in Mice (tMCAO)

tMCAO was carried out owing to previously published research [53]. Briefly, adult ICR mice (weighing 27 ± 3 g) were anesthetized with 1.5–2% isoflurane (RWD, Shenzhen, China) in a mixture of 30% oxygen and 70% nitrous oxide. The left common carotid artery, along with the internal carotid artery and the external carotid artery, were temporarily ligated. A cut was made between the two ligations on the external carotid artery. Following this, a 6-0 nylon suture (Covidien, Mansfield, MA, USA) coated with silicon (Heraeus Kulzer, Germany) was introduced through the incision and guided toward the ipsilateral middle cerebral artery. The success of middle cerebral artery occlusion was confirmed using laser Doppler flowmetry (Moor Instruments, Devon, UK) to observe a 10% reduction in the surface cerebral blood flow compared to its baseline level. The suture was withdrawn 1.5 h after the occlusion, and successful reperfusion was confirmed by observing the surface cerebral blood flow recovery back to 70% of its baseline level.

### 4.5. ASCs, ASC-CXCL12 Variant Transplantation

On the seventh day following tMCAO surgery, the mice were anesthetized and secured in a stereotaxic frame (RWD Life Science, Shenzhen, China). A solution containing 3 × 10^5^ ASCs-CXCL12, suspended in 10 μL of PBS, was carefully administered via stereotactic injection at a controlled infusion rate of 500 nl/min. The precise injection site was 2 mm lateral to the bregma and 3 mm below the dura. Subsequent to the injection, the needle was held in place for 10 min before being carefully withdrawn. Once the mice had fully recovered from the anesthesia, they were returned to their individual cages to undergo an extensive period of recuperation.

### 4.6. Administration of AMD3100

AMD3100 (Sigma) was administered intraperitoneally for 7 consecutive days starting one week after the tMCAO surgery. AMD3100 was solubilized in normal saline to achieve an injection concentration of 100 µg/mL. The dose of AMD3100 administration was 1 mg/kg/day. The control group received an equal volume of normal saline.

### 4.7. Neurological Behavioral Evaluation

To evaluate neurological function in tMCAO mice after treatment, a series of neurological scores and behavioral tests were performed using an investigator blind to the experimental groups, before and after 1 week to 5 weeks of tMCAO.

Modified Neurological Severity Scores (mNSSs) were availed to assess motor, sensory, balance, and reflex functions. The scoring scale ranged from 0 to 14, with higher scores indicating more severe neurological deficiencies and vice versa.

The rotarod test was conducted after tMCAO for the evaluation of motor function. Briefly, all animals were trained for 3 consecutive days to maintain balance and motor skills on the gradually speeding rotarod (40 revolutions per min for 2 min) before tMCAO. At 7, 14, 21, 28, and 35 days after tMCAO, the rotarod test was recorded, and the data were analyzed from 3 average trials.

The Elevated Body Swing Test (EBST) was performed to measure asymmetrical motor behavior in tMCAO mice as described elsewhere [54]. The mice were suspended vertically ~10 cm from the cage bottom by the tail, and the direction of the first upper body swing at >10° to either side of the vertical axis was recorded. Before the next swing, the tMCAO mouse was placed back in the cage to reposition. The test was repeated 20 times while resting the mice each time for about 5 min for the next swing.

The hanging wire test is designed to assess upper limb strength and coordination in mice. Once the mice were placed in the middle of the wire, the timer was started. Each mouse started with a score of 10 points, and the testing time was 180 s. The falling and reaching scores were documented during each test. A decrease in the falling score or an increase in the reaching score by 1 was recorded each time a mouse fell or reached either side of the wire. These scores were used to draw a dropping curve.

### 4.8. Ischemic Brain Infarct Volume and Atrophy Assessment

Ischemic brain infarct volume and atrophy were assessed through histological cryosections measuring 20 μm in thickness, extending from the prefrontal cortex to the hippocampus. Additionally, sequential frozen sections, 20 μm in thickness and spaced 200 μm apart, were obtained from the prefrontal cortex and were subjected to staining with 0.1% cresyl violet (Meilun Chemical Reagent Co., Dalian, China). To calculate the infarct volume, the cresyl violet-stained area in the ipsilateral hemisphere was subtracted from the corresponding area in the contralateral hemisphere. This computation was accomplished using Image J 1.52p software (NIH, Bethesda, MD, USA) and subsequently multiplied by the interval thickness of the cutting. The formula for calculating the infarct volume was as follows:(1)$$v=\sum_{1}^{n}[(s_{n}+\sqrt{s_{n}\times s_{n+1}}+s_{n+1})]\times \frac{h}{3}$$

In the given formula, “*h*” represents the interval between successive sections (0.2 mm), while “*S*” is for the measurement of the area (mm^2^) within each individual brain section.

### 4.9. Immunohistochemistry and HE Staining

The brain sections underwent a sequential treatment process, starting with exposure to 0.3% TritonX-100 for 15 min, followed by 1 h of incubation at room temperature in 1% BSA. The sections were then subjected to an overnight incubation at 4 °C with primary antibodies using the indicated dilutions, such as CD31 (1:200 dilution; R&D Systems, Minneapolis, MN, USA), rabbit-anti Ki67 (1:200 dilution, ab15580, Abcam, Cambridge, UK), and rabbit-anti DCX (Doublecortin, 1:200 dilution, ab18723, Abcam, Cambridge, UK). After three rounds of PBS washing, the sections were incubated with secondary antibodies conjugated with different fluorophores (Invitrogen) for a duration of 1 h at room temperature. Following three PBS washes, the brain slices were mounted using an antifade mounting medium containing DAPI (Invitrogen). From each mouse, images of three brain slices were taken with an FV10i confocal microscope (Olympus, Tokyo, Japan), with consistent settings throughout the image-capturing process. An appropriate scale bar was set using ImageJ for the calculation of the mean areas and the mean fluorescence integrated intensities. For H&E staining, the brain sections were fixed with 4% paraformaldehyde and stained using a H&E Kit (MB9898-3, Meilunbio, Dalian, China). Images were mosaic and merged with Leica Application Suite X.

### 4.10. Cell and Vessel Counting

For cells and vessel counting, three fields were randomly selected from the peri-infarct region using either a ×20 or ×40 objective lens. DCX^+^/Ki67^+^ cells and CD31^+^/Ki67^+^ microvessels were counted. DCX^+^ cells were located in the subventricular zone (SVZ) and CD31^+^ vessels were located in the peri-infarct area for each image (DM2500; Leica Microsystems, Wetzlar, Germany). Sections incubated with the same primary antibodies were imaged under the same conditions using an investigator blind to sample designations. The quantification of Ki67^+^, DCX^+^/Ki67^+^, CD31^+^, and CD31^+^/Ki67^+^ microvessels in the ipsilateral hemisphere was performed by an investigator who was unaware of the experimental groups. From each animal, three serial sections, spaced 400 μm apart (ranging from 1.10 mm to 0.3 mm from the bregma), were selected for analysis.

### 4.11. Western Blot Analysis

The brain tissues in the peri-infract area were collected at 3 and 5 weeks after the tMCAO. Total protein was extracted using ice-cold RIPA buffer, which contains a protease inhibitor cocktail, a phosphatase inhibitor, and phenylmethanesulfonyl fluoride. The total protein concentration was confirmed using the BCA kit (Thermo Scientific, Waltham, MA, USA). A total of 40 μg of protein from each sample was loaded into a 10% gel for electrophoresis. The proteins were electrophoretically transferred to a PVDF membrane, which was then blocked with a 10% protein-free protein-blocking solution (Epizyme, Shanghai, China) for 15 min at room temperature. Subsequently, the membrane was incubated with primary antibody, including VEGF (1:1000, rabbit, Abcam), bFGF antibody (1:1000, 05–118, Millipore, Billerica, MA, USA), and *β*-actin (1:1000, mouse, Invitrogen) at 4 °C overnight.

Following the primary antibody incubation, the membrane was incubated with the corresponding secondary antibodies, namely, horseradish peroxidase (HRP)-conjugated goat anti-rabbit IgG or goat anti-mouse IgG, for one hour at 37 °C. After being washed thrice with TBST, the membrane was placed in an imaging system and treated with an enhanced chemiluminescence substrate. Subsequently, the images were collected and analyzed using ImageJ 1.52p software.

### 4.12. Real-Time PCR Analysis

Total RNA was isolated using TRIZOL Reagent (Invitrogen) according to the manufacturer’s protocol. The concentration of isolated RNA was examined using a spectrophotometer (NanoDrop1000, Thermo, Wilmington, DE, USA). cDNA was synthesized from the isolated RNA using the cDNA Reverse Transcriptase Kit (Yeasen, Shanghai, China). The primer used for the target genes was as follows:GAPDH:Forward: CTGGGCTACACTGAGCACC;Reverse: AAGTGGTCGTTGAGGGCAATG,VEGF:Forward: CTGCCGTCCGATTGAGACC;Reverse: CCCCTCCTTGTACCACTGTC,bFGF:Forward: GCGACCCACACGTCAAACTA;Reverse: TCCCTTGATAGACACAACTCCTC,CXCR4:Forward: GACTGGCATAGTCGGCAATG;Reverse: AGAAGGGGAGTGTGATGACAAA,CXCL12:Forward: TGCATCAGTGACGGTAAACCAReverse: CACAGTTTGGAGTGTTGAGGAT.

The real-time PCR reaction mix was prepared with a concentration of 1X SYBER Green, 200 nM gene-specific primers, and an equal amount of cDNA. The PCR reaction was performed under the following conditions: 95 °C for 30 s followed by 40 cycles at 95 °C for 5 s and 60 °C for 30 s. To calculate mRNA levels, the Δ/Δ Ct method was employed, where we normalized the relative mRNA expression to that of GAPDH mRNA in the same samples. These steps were performed to analyze the mRNA expression levels of the target genes (VEGF, bFGF, CXCR4, and CXCL12) and to compare their expression in different experimental conditions.

### 4.13. Statistical Analysis

All the data in this project were analyzed using GraphPad Prism version 9.0 and presented as mean ± SD. The statistical test of one-way ANOVA was used for the comparison of multiple treatment groups. The comparisons of means between the two groups were analyzed using a two-tailed Student’s *t* test. A *p* value < 0.05 was considered statistically significant.

## 5. Conclusions

Our study highlights the remarkable potential of mouse-derived ASC transplantation, engineered with the monomeric CXCL12 variant, in enhancing neurobehavioral recovery following ischemic stroke. This therapeutic effect is attributed to the increased stimulation of angiogenesis and neurogenesis, thereby highlighting the significance of utilizing ASC-mCXCL12 for the treatment of stroke.

## Figures and Tables

**Figure 1 ijms-25-00792-f001:**
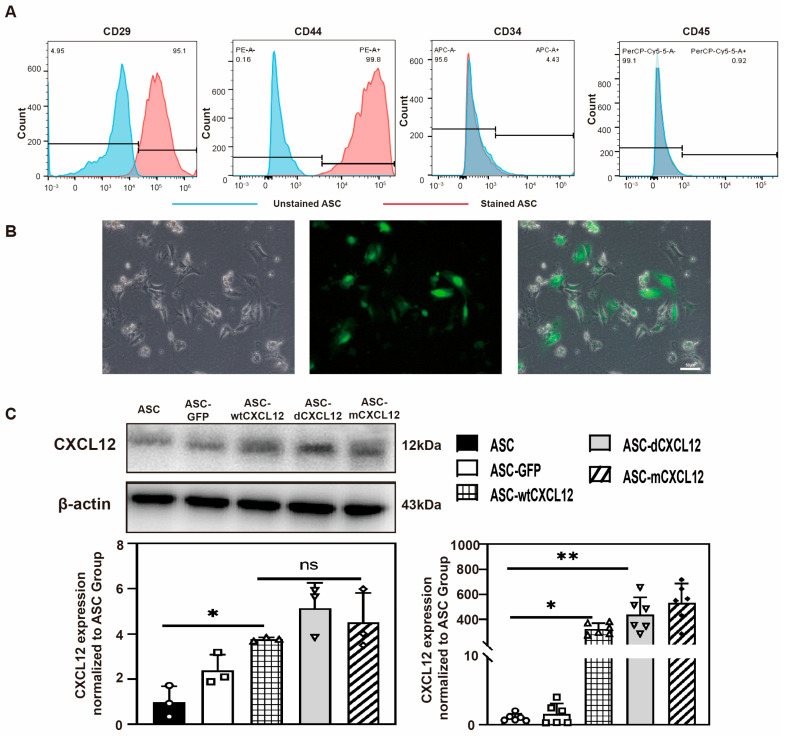
Characterization of ASC and CXCL12 overexpression in ASCs: (**A**) Flow cytometric analysis of ASC surface markers, CD29 and CD44, and ASC negative markers CD34 and CD45. (**B**) CXCL12-GFP transfected ASCs in a bright field (left), fluorescent field (middle), and merged (right). Scale bar = 50 μm. (**C**) Western blotting analysis and real-time PCR to detect CXCL12 protein (left) and mRNA (right) expression in ASCs, ASC-GFP, ASC-wtCXCL12, ASC-dCXCL12, and ASC-mCXCL12, *n* = 3 per group. The data are presented as the mean ± SD. * *p* < 0.05, ** *p* < 0.01. ns = not significant.

**Figure 2 ijms-25-00792-f002:**
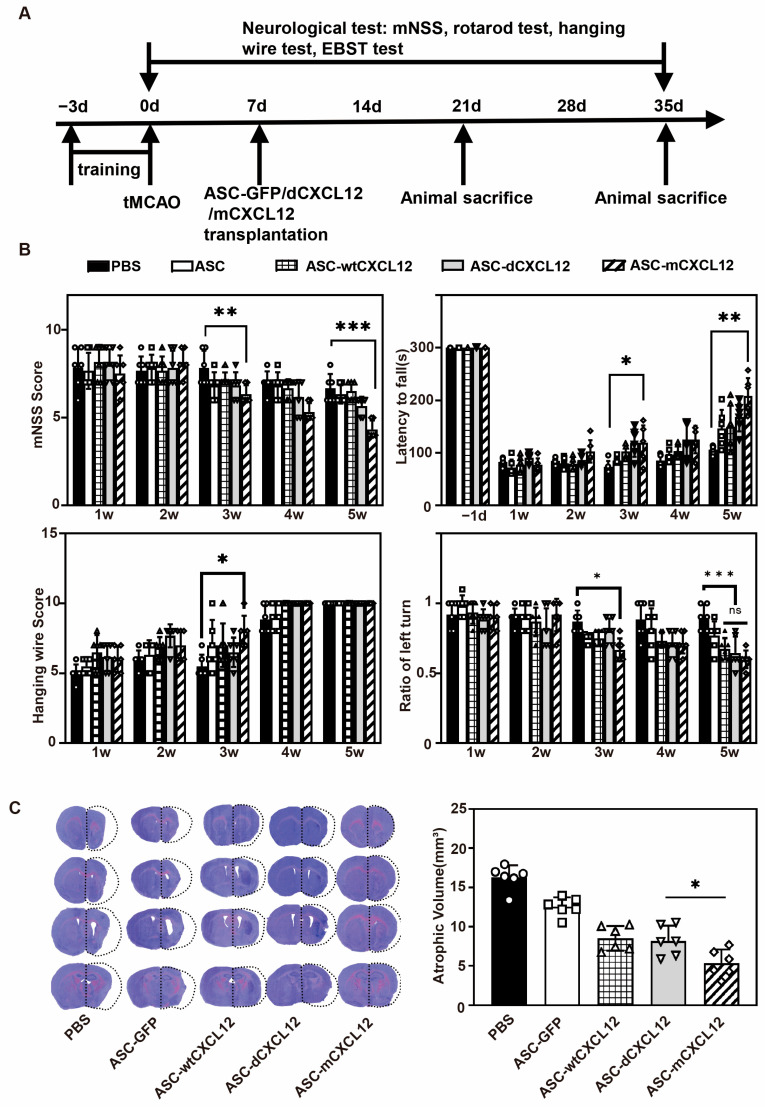
The transplantation of mCXCL12-ASCs improved neurobehavioral outcomes and reduced brain atrophy at 5 weeks after tMCAO. (**A**) Experimental schedule. An mCXCL12-ASC injection was performed 7 days after tMCAO. (**B**) mNSS, rotarod, hanging wire, EBST evaluation in PBS, ASC, ASC-wtCXCL12, ASC-dCXCL12, and ASC-mCXCL12 groups. There were 6–8 mice per group. (**C**) Cresyl violet-stained brain slice for brain atrophy evaluation after 5 weeks of tMCAO (left). Brain atrophy volume quantification (right). There were 5–6 mice per group. Data presented as mean ± SD.* *p* < 0.05, ** *p* < 0.01, *** *p* < 0.001.

**Figure 3 ijms-25-00792-f003:**
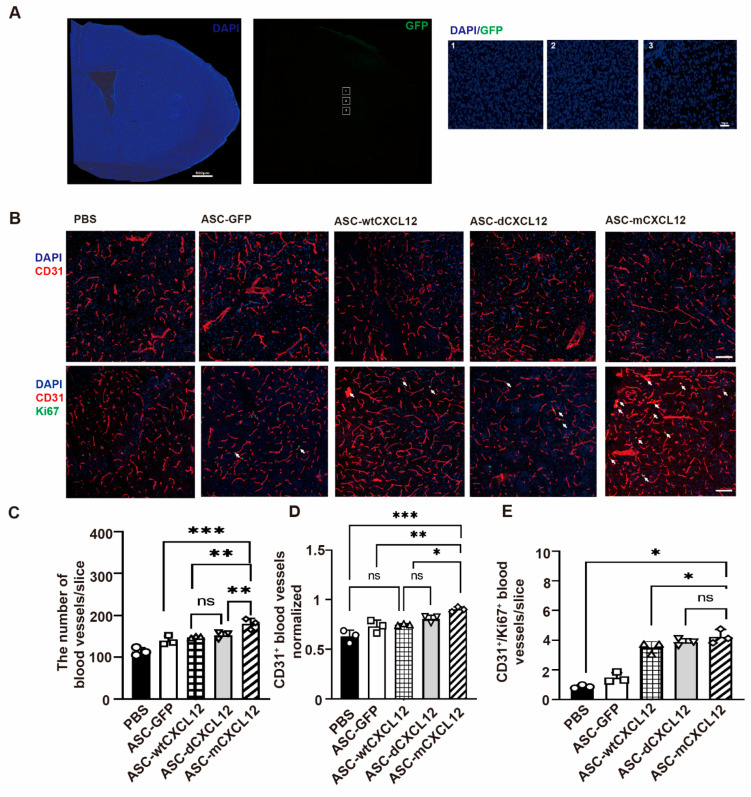
Immunohistochemical analysis of angiogenesis in mice after tMCAO: (**A**) Immunostaining images of ipsilateral brain slices (left panel) from ASC-GFP group 3 weeks after tMCAO; scale bar = 500 μm. Representative images (right panels) from the white squared zone. Scale bar = 50 μm. (**B**) Images of the peri-infarct area 5 weeks after tMCAO in the control, ASC, ASC-wtCXCL12, ASC-dCXCL12, and ASC-mCXCL12 groups. CD31^+^ (red)/Ki67^+^ (green) staining shows the cells in the peri-infarct area 5 weeks after tMCAO. White arrows indicated Ki67+/CD31+ cells. Scale bar = 100 μm. (**C**) Quantitative analysis of the number of blood vessels in the control, ASC, ASC-wtCXCL12, ASC-dCXCL12, and ASC-mCXCL12 groups 5 weeks after tMCAO. (**D**) Quantitative analysis of the number of blood vessels normalized to the contralateral field in the control, ASC, ASC-wtCXCL12, ASC-dCXCL12, and ASC-mCXCL12 groups 5 weeks after tMCAO. (**E**) Quantification of CD31^+^/Ki67^+^ cells in mice from control, ASC, ASC-wtCXCL12, ASC-dCXCL12, and ASC-mCXCL12 groups 5 weeks after tMCAO. Five mice per group. Data presented as mean ± SD. * *p* < 0.05, ** *p* < 0.01, *** *p* < 0.001, ns = not significant.

**Figure 4 ijms-25-00792-f004:**
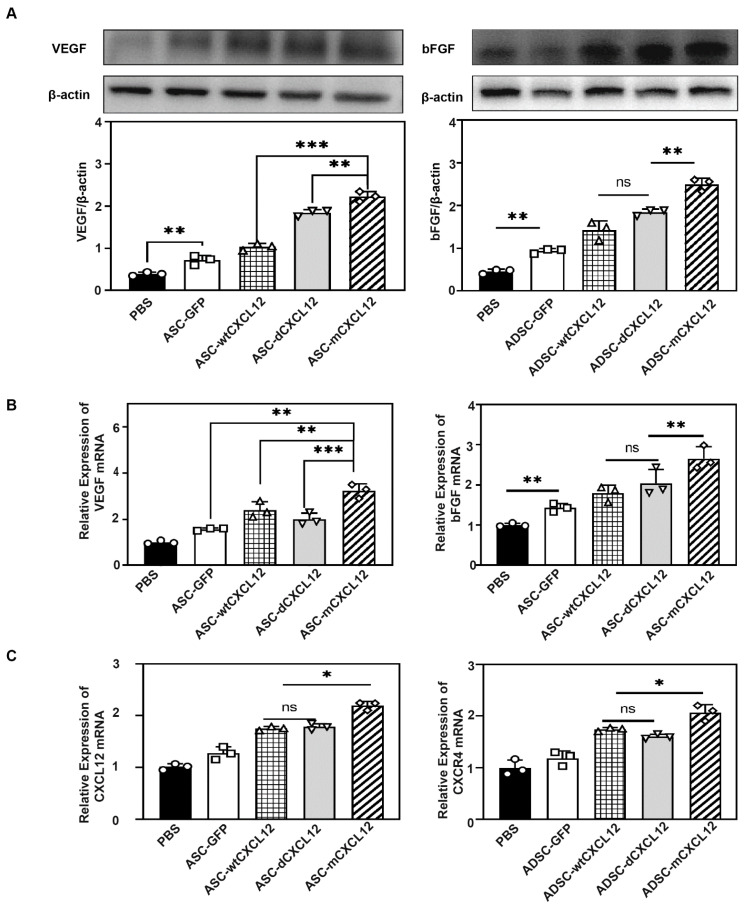
Expression of VEGF and bFGF in mouse brain after tMCAO. (**A**) Western blot for the evaluation of VEGF and bFGF expression 5 weeks after tMCAO. Quantitative analysis of VEGF/β-actin (left), bFGF/β-actin (right) in the control, ASC, ASC-wtCXCL12, ASC-dCXCL12, and ASC-mCXCL12 groups 5 weeks after tMCAO. (**B**) Real-Time PCR to detect the expression of VEGF (left) and bFGF (right) in the control, ASC, ASC-wtCXCL12, ASC-dCXCL12, and ASC-mCXCL12 groups 5 weeks after tMCAO. (**C**) Real-time PCR to detect the expression of CXCL12 (left) and CXCR4 (right) in the control, ASC, ASC-wtCXCL12, ASC-dCXCL12, and ASC-mCXCL12 groups 5 weeks after tMCAO. Five mice per group. Data presented as mean ± SD. * *p* < 0.05, ** *p* < 0.01, *** *p* < 0.001, ns = not significant.

**Figure 5 ijms-25-00792-f005:**
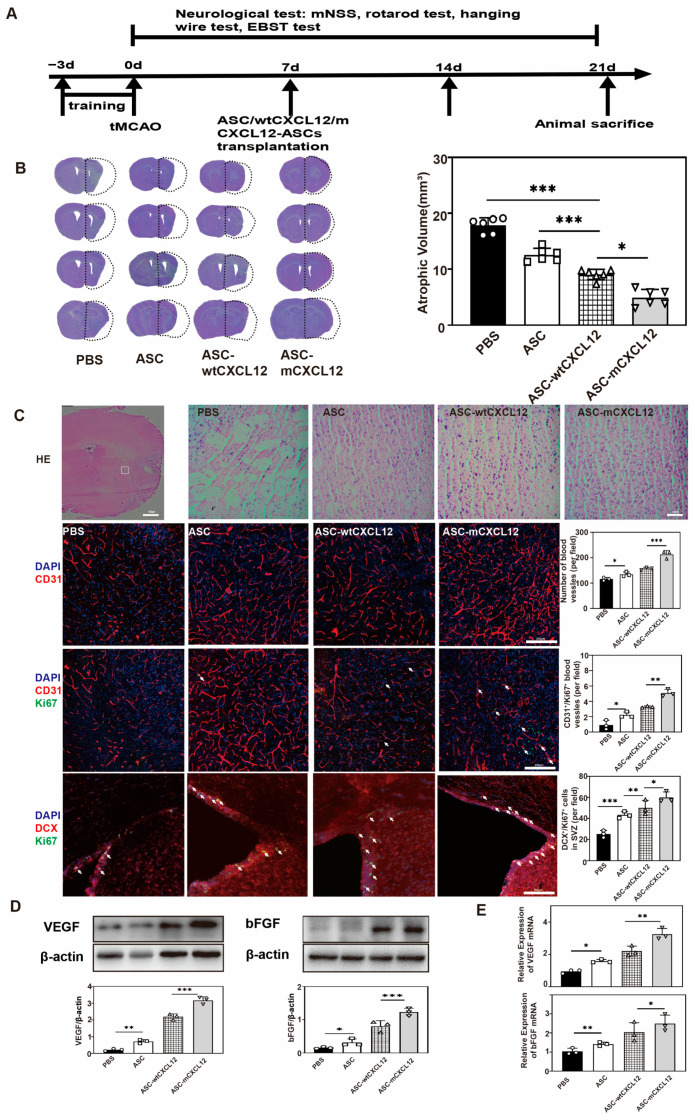
ASC-mCXCL12 transplantation improved angiogenesis and neurogenesis of the mouse brain 3 weeks after tMCAO. (**A**) Experimental schedule. ASC-mCXCL12 injection was performed 7 days after tMCAO. (**B**) Cresyl violet-stained brain slice for brain atrophy evaluation after 3 weeks of tMCAO (left). Brain atrophy volume quantification in the control, ASC, ASC-wtCXCL12, and ASC-mCXCL12 groups at 3 weeks after tMCAO (right). (**C**) H&E image of ipsilateral brain slices from ASC group at 3 weeks after tMCAO; scale bar = 500 μm. Representative images from control, ASC, ASC-wtCXCL12, and ASC-mCXCL12 groups; scale bar = 50 μm. Immunostaining images in the peri-infarct area after tMCAO. Red color shows CD31^+^ staining, and green shows Ki67^+^ (middle) and DCX^+^ (bottom) staining. White arrows indicated Ki67+/CD31+ cells (middle) and Ki67+/DCX+ cells (bottom). Quantitative analysis of the number of CD31^+^, CD31^+^/Ki67^+^, and DCX^+^/Ki67^+^ per field in the control, ASC, ASC-wtCXCL12, and ASC-mCXCL12 groups. *n* = 3, 3, 3, 3. Scale bar = 200 μm. (**D**) Quantitative analysis of VEGF/β-actin (left), bFGF/β-actin (right) in PBS, ASC, ASC-wtCXCL12, and ASC-mCXCL12 groups 3 weeks after tMCAO. (**E**) Real-time PCR to detect the expression of VEGF and bFGF in PBS, ASC, ASC-wtCXCL12, and ASC-mCXCL12 groups 3 weeks after tMCAO. Five mice per group. Data presented as mean ± SD. * *p* < 0.05, ** *p* < 0.01, *** *p* < 0.001.

**Figure 6 ijms-25-00792-f006:**
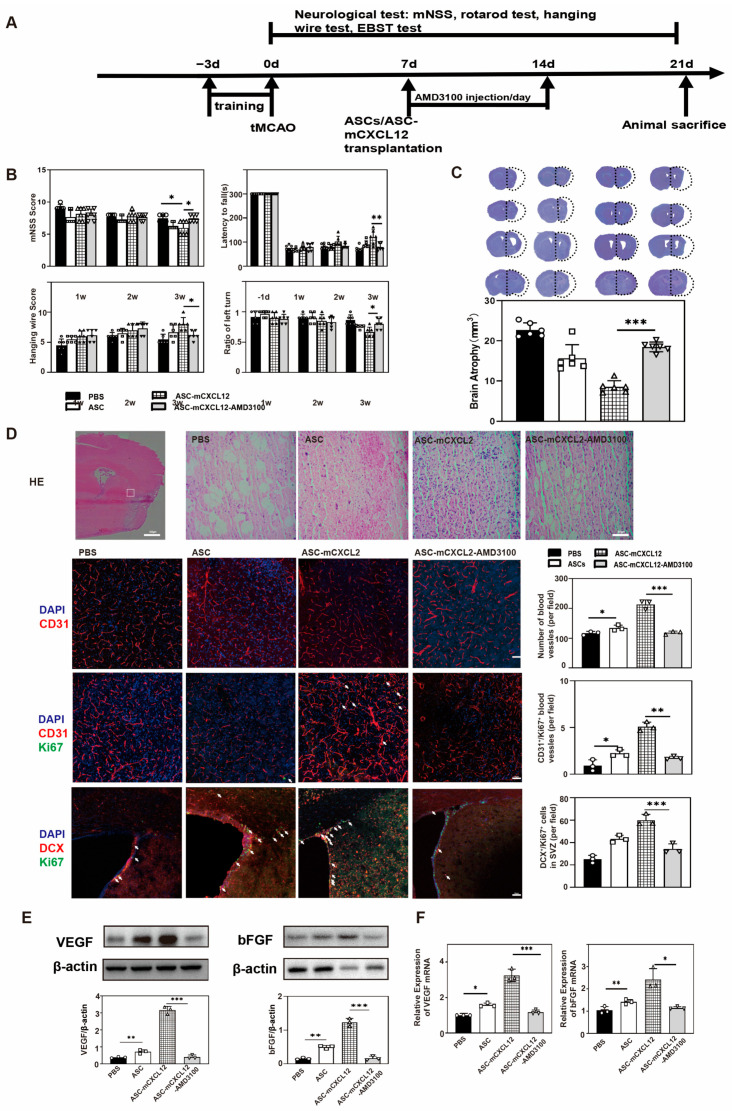
ASC-mCXCL12 functions in ischemic mice are mediated using the CXCL12/CXCR4 signaling axis: (**A**) Experimental schedule. ASC-mCXCL12 and AMD3100 injections were performed 7 days after tMCAO. (**B**) mNSS, rotarod, hanging wire, and EBST evaluation in PBS, ASC, ASC-mCXCL12, and ASC-mCXCL12-AMD3100 groups. There were 6–8 mice per group. (**C**) Cresyl violet-stained brain slice for brain atrophy evaluation after 3 weeks of tMCAO (left). Brain atrophy volume quantification in the control, ASC, ASC-mCXCL12, and ASC-mCXCL12-AMD3100 groups 3 weeks after tMCAO (right). (**D**) H&E image of ipsilateral brain slices from ASC group 3 weeks after tMCAO; scale bar = 500 μm. Representative images from control, ASC, ASC-mCXCL12, and ASC-mCXCL12-AMD3100 groups; scale bar = 50 μm. Immunostaining images in the peri-infarct area after tMCAO. The red color represents CD31^+^ staining, and green shows Ki67^+^ (middle) and DCX^+^ (bottom) staining. White arrows indicated Ki67+/CD31+ cells (middle) and Ki67+/DCX+ cells (bottom). Quantitative analysis of the number of CD31^+^, CD31^+^/Ki67^+^, and DCX^+^/Ki67^+^ per field in the control, ASC, ASC-mCXCL12, and ASC-mCXCL12-AMD3100 groups. Scale Bar = 50 μm. (**E**) Quantification of VEGF/β-actin (left) and bFGF/β-actin (right) in the control, ASC, ASC-mCXCL12, and ASC-mCXCL12-AMD3100 groups at 3 weeks after tMCAO. (**F**) Real-time PCR to detect the expression of VEGF and bFGF in the control, ASC, ASC-mCXCL12, and ASC-mCXCL12-AMD3100 groups 3 weeks after tMCAO. Five mice per group. Data presented as mean ± SD. * *p* < 0.05, ** *p* < 0.01, *** *p* < 0.001.

**Figure 7 ijms-25-00792-f007:**
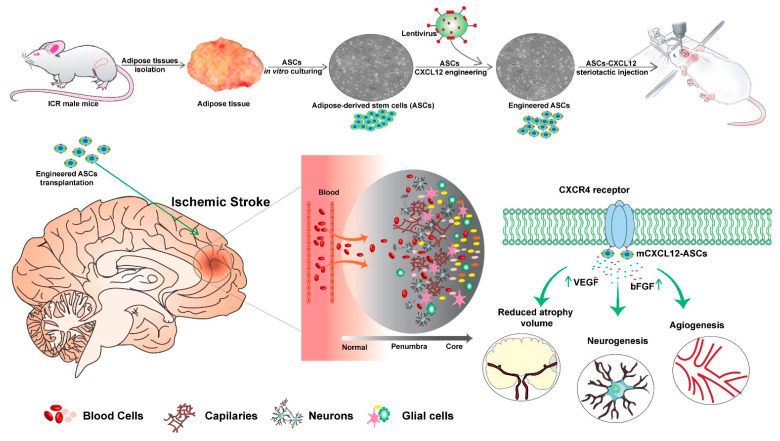
The therapeutic effect of ASC-mCXCL12 after tMCAO. This diagram illustrates that ASCs derived from adult ICR mice engineered with monomeric CXCL12 were implanted into the peri-infarct area of tMCAO mice via stereotactic injection. The results demonstrated that ASC-mCXCL12 treatment could increase neurogenesis and angiogenesis and reduce atrophy volume. This outcome is likely attributed to the interaction of monomeric CXCL12 with CXCR4, which regulates the recruitment of bFGF and VEGF.

## Data Availability

Data is contained within the article.

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
