# Peer review of "Monomeric CXCL12-Engineered Adipose-Derived Stem Cells Transplantation for the Treatment of Ischemic Stroke"

_ijms, 2024, doi:10.3390/ijms25020792_

Round 1

Reviewer 1 Report

Comments and Suggestions for Authors

In the present study, authors applied lentivirus-based gene delivery to over-expression different forms of CXCL12 in ADSCs, followed by ADSC transplantation on mice with tMACO. In addition to behavior observation, histopathological examination, immunofluorescent stainings, Western blots as well as RT-PCR were performed to evaluate the effect of engineered CXCL12-ADMSC on neural functional preservation and cerebral angiogenesis of mice with ischemic stroke. Although this is an interest and meaningful study in the field of cell therapy in ischemic stroke, crude data collection and poor manuscript preparation limit its importance. Besides, there are several points needed to be clarified. 

1.     In most studies, due to the severe cerebral necrosis and irreversible fibrosis after brain ischemic injury, the application of stem cells on tMCAO mouse mode was utilized within 48 hours post induction of ischemic stroke. However, ADSCs transplantation was applied 7 days after tMCAO in this study. It is concerned whether the pro-angiogenic effect of ADSC could have a functional benefit in the brain with glial scar.

2.     Nuclear DAPI counter-staining is necessary for all immunofluorescent stainings. Moreover, high-resolution histological staining, such as H.E. staining, should be included in Fig. 5 and 6.

3.     The validation of transplanted ADSCs in the ischemic cerebral cortex should be provided. According the description on section 2.2, the lentivirus for CXCL12 overexpression should co-express GFP via IRES. Hence, images for tracking transplanted ADSCs by different days after transplantation through GFP signal observation should be included in Fig.2. Moreover, how authors distinguish fluorescent signals from lenti-GFP and green fluorescence-conjugated 2nd antibodies in Fig. 3, 5, and 6. 

4.     Where is the location for ADSCs transplantation, striatum or ventricle? The coordinates for stereotaxic injection should be provided (AP, ML, DV). 

5.     The quality of Western blottings and quantitation should be improved. In Fig. 1C, the density of actin expression in ADSC-mCXCL12 is significantly higher than in other groups. The quantitation and calculation for CXCL12 expression for this experiment may not be accurate. Moreover, the selection of images presented in each figure should be labeled in supplementary original blots. For example, the blot for actin expression in Fig. 1C could not be found in the original images.

6.     The specificity of antibodies should be concerned. In original blot of Fig. 1C, lots of unfavored intense bands in CXCL12 Western blotting. Moreover, the blotting patterns of VEGF and bFGF are distinct in Fig. 5 and 6. 

7.     For statistics, in the last sentence of the legend of Fig. 1, what is the meaning of “ns *p<0.05”? Moreover, it is hard to understand the labeled symbols for statistics. Are they indicated for the comparison with ADSC only or PBS group? What’s the method for inter-group comparisons post ANOVA?

8.     What’s the full name of DCX in Fig. 5? Doublecortin?

9.     English editing is necessary for the abstract.    

Reviewer 2 Report

Comments and Suggestions for Authors

The article explores the potential of monomeric CXCL12-modified adipose-derived stem cells (ADSCs) as a novel approach for improving stroke treatment. The study involves the introduction of monomeric, dimeric, or wild-type CXCL12 genes into ADSCs using lentiviral vectors. The results demonstrate that stereotactic transplantation of ADSCs modified with monomeric CXCL12 effectively promotes neurobehavioral recovery in mice by stimulating angiogenesis and neurogenesis. However, there are some major limitations in this study:

- Histological analysis is missing. Ideally a figure on cell survival and differentiation into neural cell types should be shown for each group. What proportion of the injected stem cells become mature neurons, or do they stay in a 'stem' state? 

-          Please show individual data points for all bar plots in all figures where n <10, to show data distribution more clearly.

-          In methods, it says n=20 per group. But for behavioral analysis, only 6-8 mice are included per group? Please explain and provide rationale in manuscript why other mice were excluded.

-          mCXCl12 group shows extremely low levels of brain atrophy, including minimal enlargement of ventricles that is commonly seen after tMCAOs. However, figure legends shows that this analysis was done on only 25% (5/20 mice) of the total mice per group? n=5 is too low for this type of measurements. This analysis needs to be done on all mice from each group.

-          tMCAo strokes results in lesions of various sizes and locations due to collateral blood flow. Where are the peri-infarct images taken for angiogenesis measurements? Are they taken at the same brain region in each mouse? Or different region in each mouse depending on the location of lesion in that mouse? Ideally, you would use the same peri-infarct region where the cells were implanted. 

- On a similar note, since the lesion size/location should be different for each mouse, does the survival of the implanted cells depend on the distance from the lesion? How does this distance correlate with behavioral performance? 

-          Figure 1A: please label each plot with the name of marker (ex: CD29+, CD44+ etc) so it is clear without having to look through the text.

-          While the discussion touches on the role of CXCR4 and CXCL12 in the study, it could benefit from a more detailed exploration of the underlying mechanisms. Explain how monomeric CXCL12 interacts with CXCR4 and how this interaction influences neurobehavioral recovery.

-          It's important to acknowledge the limitations of the study and how they may affect the interpretation of the results. Discuss any potential sources of bias or confounding factors that might have influenced the outcomes.

Reviewer 3 Report

Comments and Suggestions for Authors

The Authors would like to show the therapeutic efficacy of ASCs in the treatment of ischemic stroke by means a genetic engineering of those cells with a monomeric structure of CXCL12.

The article presents several pitfalls and in particular:

-the terminology of adipose stem cells is ASCs and not ADSC;

-the origin of ASCs must be present either In the Title (if human hASCs) and in the Abstract;

-In the Introduction a brief description of ASCs must contain the several aspects including their main activities.

-In the Introduction, the description of CXCl12 activities not correlated with this study must be deleted;

-In Mat and Met the TIMES of use of cells and of recovery must be clarified. The experimental design must be better described and a GRAPHICAL abstract of the experiments is needed (add a Figure);

-The Results are critical regarding the Figure 1 B and C. Fig. 1 C in the merge that is opaque and, consequently, the positivity is lower than in the IF image.

In this figures the blots must be substituted- the beta actin figure has been altered maybe only in its magnification and cannot be accepted.

Also in other cases the RT-PCR and WB figures are critical, therefore the Authors must show ALL the originals;

- In particular the Figure 1 C is not acceptable as in it the standard (beta actin) is differently loaded and the CXCL12 is almost not detectable;

-the number of dead animals is not reported as well as the side effects in particular. The total number is very high.

-Discussion is too long and must be shortened;

-A para with the Limitations of this study must be added at the end of the Article before the Conclusions.

Comments on the Quality of English Language

English must be revised.

Round 2

Reviewer 2 Report

Comments and Suggestions for Authors

Thank you for your comments. The authors made appropriate changes to their manuscript. I have one comment:

The authors mention that they did not find any GFP+ cells in the brain 3 weeks post-injection. Since the cells were transfected with GFP prior to injection, these cells should not lose GFP signal even if differentiated to other cell types (unless the GFP signal is cell type reporter specific which does not seem to be the case here). This suggest that the injected cells actually did not survive for 3 weeks. This is a major point that needs to be highlighted in the manuscript. 

Please describe in your manuscript that these mice had no transplanted cells surviving at 3 weeks, and also please explain in detail that your effects are not from cell replacement but potentially from signaling factors released while cells were alive. 

Reviewer 3 Report

Comments and Suggestions for Authors

The Authors have answered to the majority of my previous concerns and the paper is not much better and clear.

A few comments must still be addressed:

1) Being the ASCs belonging to mice and not to humans the translational ability of this study is much lower. It is very easy to obtain ASCs from humans and this reviewer cannot understand why the Authors used mice ASCs. The limits must be added in the Conclusions/Limits of this study and it must be clearer in the whole manuscript what is the origin of those cells;

2) the blots of VEGF sent (originals) of Figure 4 A1 are not available for any presentation because two bands are overlapped;

Comments on the Quality of English Language

Minor spelling check
